# Machine learning identifies pupil size and corneal thickness as key predictors of axial elongation rate

Peng Zhou[1,2], Sitong Chen[1,3,4,5,6], Yingli Li[1,3,4,5,6], Yan Li[1,3,4,5,6]*

1 Optometry Center, Peking University People's Hospital, Beijing, China, 2 Department of Ophthalmology, ParkwayHealth Shanghai, Shanghai, China, 3 Department of Ophthalmology, Peking University People's Hospital, Beijing, China, 4 Beijing Key Laboratory of Diagnosis and Therapy of Ocular Diseases and Optometry, Beijing, China, 5 School of Optometry and Ophthalmology, Peking University Health Science Center, Beijing, China, 6 Institute of Medical Technology, Peking University Health Science Center, Beijing, China

* eyedrliyan@outlook.com

## Abstract

### Purpose

This study aimed to develop a machine learning-based prediction model for myopia progression using ocular biometric parameters to provide an objective assessment tool for clinical practice.

### Methods

A retrospective analysis was conducted on patients treated at Shanghai Parkway Health Ophthalmology Department as the training set, and myopic individuals from the Optometry Center of Peking University People's Hospital as the validation set. Demographic and biometric data were collected, including central corneal thickness (CCT), axial length (AL), corneal curvature (K-value), anterior chamber depth (ACD), corneal diameter (WTW), and pupil size (PS). Seven machine learning models (e.g., XGBoost, random forest, support vector machine) were employed for modeling, with performance optimized via 5-fold cross-validation. Model accuracy was evaluated using mean squared error (MSE) and the coefficient of determination (R²), and variable importance was analyzed.

### Results

No statistically significant differences were observed in baseline characteristics between the training and validation sets (all P > 0.05). The XGBoost model demonstrated the best performance, achieving R² = 0.913 (MSE = 0.005) on the training set and R² = 0.766 (MSE = 0.016) on the test set. Variable importance analysis revealed pupil size (score 100) and corneal thickness (40.88) as the key predictors of axial elongation rate, followed by age of onset (17.96).

**Data availability statement:** All relevant data are within the manuscript and its Supporting Information files.

**Funding:** The author(s) received no specific funding for this work.

**Competing interests:** The authors have declared that no competing interests exist.

## Conclusion

The machine learning-based prediction model effectively utilizes ocular biometric data to assess myopia progression risk, with pupil size and corneal thickness identified as core predictive factors. This model provides a quantitative tool for early clinical intervention. Future studies should expand the sample size and incorporate additional biomarkers to optimize performance.

---

## Introduction

Myopia has emerged as a major global public health concern [1,2]. In East Asia, the prevalence of myopia among adolescents reaches 53%–96% [3]. High myopia may lead to sight-threatening complications such as cataracts, choroidal neovascularization, retinal detachment, glaucoma, and macular atrophy [4,5], and even irreversible vision loss or blindness. Therefore, controlling myopia progression is of critical importance.

Risk factor analysis can help identify children at high risk of myopia onset or rapid progression, enabling early targeted interventions to delay onset or slow progression [6]. Our previous research examined ocular biometric parameters, including central corneal thickness (CCT), axial length (AL), corneal curvature (K-value), anterior chamber depth (ACD), corneal diameter (WTW), and pupil size (PS), and identified a significant correlation between CCT and myopia progression [7]. However, our prior study relied on linear regression, which has limitations in data mining depth.

Compared to traditional statistical methods, machine learning (ML) techniques offer distinct advantages in exploring variable relationships [8]. Their core value lies in uncovering complex, nonlinear interactions among multidimensional datasets while constructing high-precision predictive models [9]. Notably, ML algorithms overcome the restrictive assumptions of linearity and homoscedasticity required by conventional approaches, allowing the detection of latent variable interactions and nonlinear associations [10].

This study employs ML algorithms to develop a predictive model that estimates myopia progression risk based on biometric data. The goal is to provide clinicians and optometrists with an objective tool to assess progression trends in young myopic patients using measurable ocular parameters.

## Materials and methods

### Study framework and participant recruitment

This study is a retrospective analysis.We utilized patient data collected from Shanghai Parkway Health between January 1, 2020 to January 31, 2022 as the training set, with myopic patients from the Ophthalmology and Optometry Center of Peking University People#39;s Hospital serving as the validation cohort from January 1, 2024 to May 31, 2025. Only right eye measurements were analyzed for all participants. The inclusion criteria comprised: (1) aged 4−18 years; (2) spherical equivalent refraction ≤ −0.5D in right eye after cycloplegia; (3) absence of other severe ocular

pathologies; (4) written informed consent obtained from both participants and their legal guardians after full explanation of the study purpose and procedures; and (5) minimum follow-up duration of 12 months. All children included in this study were managed without any myopia control interventions (myopia control glasses, multifocal contact lenses, etc.) and wore only single-vision corrective lenses. Age of onset is defined as the age at which myopia was first diagnosed and corrective spectacles were prescribed.

The data were accessed for research purposes on May 1, 2025. Authors did not have access to information that could identify individual participants during or after data collection.

### Ethics statement

The study complied with the ethical standards set forth by the Declaration of Helsinki. The research protocol was approved by the Institutional Review Board of Peking University People's Hospital (Approval No.: 2021PHB322−001). Given the retrospective design and the use of anonymized data, the IRB granted a waiver of informed consent for this study.

### Examinations

This study implemented a standardized examination protocol across both ophthalmology centers, with professional ophthalmologists evaluating the anterior segment and fundus of each participant using a slit-lamp system. Each subject underwent two cycloplegic refraction measurements for both eyes. Cycloplegia was induced with three instillations of compound tropicamide (0.5% tropicamide and 0.5% phenylephrine hydrochloride; Santen Pharmaceutical, Japan), administered at 5-minute intervals. Refraction was measured 30 minutes after the final dose, following confirmed mydriasis.Baseline refraction was measured using a Topcon KR-8800 autorefractor (Topcon Corporation, Japan), with follow-up measurements taken after 12 months. Cycloplegic refraction measurements were used for all analyses in this study. Additionally, ocular biometric parameters—including central corneal thickness (CCT), axial length (AL), corneal curvature (K-values), anterior chamber depth (ACD), white-to-white corneal diameter (WTW), and pupil size (PS)—were obtained using the AL-Scan optical biometer (Nidek Co., Japan). Spherical equivalent refraction (SE) was calculated using the standard formula: sphere power plus half of the cylinder power. Ocular biometry and baseline refractive assessments were performed by multiple trained examiners, each with over ten years of clinical experience, ensuring consistency and reliability of the measurements.. Throughout the study, no interventions—such as orthokeratology lenses or low-dose atropine eye drops—were administered for myopia control.

### Development and application of the prediction model

This study incorporated multiple predictive variables, including age of onset, corneal curvature, anterior chamber depth, corneal thickness, corneal diameter, pupil size, and axial elongation rate. To ensure data integrity, missing values were excluded, and all independent variables underwent standardization (centering and min-max normalization). The standardization model was constructed based on the training dataset and subsequently applied to the testing dataset. Using stratified sampling, the data were partitioned into training (70%) and testing (30%) sets, ensuring a balanced distribution of the dependent variable (axial elongation rate) across both subsets. Stratified sampling was performed based on axial elongation rate tertiles to ensure balanced distribution.

Furthermore, seven machine learning algorithms were employed for regression modeling of axial elongation rate: Random Forest (RF), Extreme Gradient Boosting (XGBoost), Support Vector Machine (SVM), Neural Network (NNet), Regression Tree, Ridge Regression, and Lasso Regression. All models were trained and optimized using 5-fold cross-validation, repeated 10 times for robustness. Hyperparameter tuning was performed with appropriate search grids: RF was constrained to a maximum of 10 terminal nodes, XGBoost utilized a fixed learning rate of 0.05, SVM adopted a radial basis

function (RBF) kernel, and the neural network model was tested with three distinct combinations of node sizes and decay parameters. Both Ridge and Lasso regression employed an automated tuning grid with a length of 5.

Moreover, model performance was evaluated on both training and testing sets using mean squared error (MSE) and the coefficient of determination ($R^2$). The model achieving the highest $R^2$ on the testing set was selected as the "optimal model" for subsequent variable importance analysis.

## Statistical analysis

Statistical analyses were performed using R software (version 4.3). All accuracy analyses of different predictive models in this study were processed by R software. All independent variables underwent standardization. The measurement data of the assessed indicators passed the Kolmogorov-Smirnov (K-S) test for normal distribution and are presented as mean ± standard deviation (SD). A P-value < 0.05 was considered statistically significant.

## Results

### Patient characteristics

The training cohort comprised 259 patients from the Ophthalmology Department of Shanghai Parkway Health, while the validation cohort consisted of 109 myopic individuals from the Ophthalmology and Optometry Center of Peking University People's Hospital. Key baseline characteristics, including age of onset, corneal curvature, central corneal thickness, corneal diameter, anterior chamber depth, pupil size, myopia progression rate, and axial elongation rate, are presented in Table 1. Comparative analysis demonstrated no statistically significant differences in these parameters between the two cohorts (all *P* > 0.05).

### Model performance comparison

The predictive performance of the seven regression models on both training and test sets is summarized in Table 2. Overall, the XGBoost model demonstrated superior performance, achieving the highest coefficient of determination ($R^2$ = 0.913) and lowest mean squared error (MSE = 0.005) on the training set. This advantage was maintained in the test set ($R^2$ = 0.766, MSE = 0.016), outperforming all other evaluated models. Consequently, XGBoost was selected as the optimal model for subsequent interpretative analyses.

### Variable importance and SHAP analysis

The variable importance analysis of the XGBoost model is presented in Fig 1. Pupil size emerged as the most influential factor affecting axial elongation rate, with the highest importance score (100), followed by corneal thickness (40.88) and age of onset (17.96). In contrast, anterior chamber depth, corneal diameter, and corneal curvature

**Table 1. Patient characteristics in the training and validation sets.**

| Parameter | Training set | Validation set | Intergroup difference |
|---|---|---|---|
| Age of onset (years) | 9.19 ± 2.58 | 9.35 ± 2.41 | p > 0.05 |
| Corneal curvature (D) | 43.22 ± 1.53 | 43.59 ± 1.75 | p > 0.05 |
| Central corneal thickness (µm) | 543.74 ± 29.24 | 549.00 ± 20.20 | p > 0.05 |
| Corneal diameter (mm) | 11.97 ± 0.39 | 11.61 ± 0.52 | p > 0.05 |
| Anterior chamber depth (mm) | 3.83 ± 0.25 | 3.65 ± 0.30 | p > 0.05 |
| Pupil size (mm) | 3.52 ± 0.47 | 3.61 ± 0.79 | p > 0.05 |
| Myopia progression rate (D/year) | 0.89 ± 0.40 | 0.79 ± 0.42 | p > 0.05 |
| Axial length progression rate (mm/year) | 0.65 ± 0.17 | 0.55 ± 0.17 | p > 0.05 |

**Table 2. Predictive Performance of Different Machine Learning Models on Training and Test Sets.**

| Model | Training set | | Validation set | |
|---|---|---|---|---|
| | MSE | R² | MSE | R² |
| rf | 0.011 | 0.826 | 0.019 | 0.733 |
| xgb | 0.005 | 0.913 | 0.016 | 0.766 |
| svm | 0.020 | 0.670 | 0.022 | 0.680 |
| nnet | 0.019 | 0.665 | 0.023 | 0.658 |
| tree | 0.010 | 0.820 | 0.020 | 0.709 |
| ridge | 0.020 | 0.663 | 0.023 | 0.657 |
| lasso | 0.020 | 0.661 | 0.023 | 0.666 |

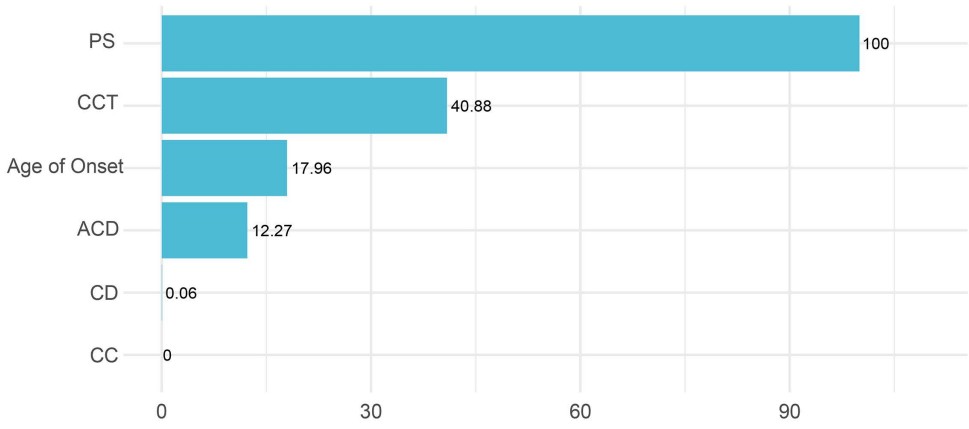

**Fig 1. Variable importance analysis of the XGBoost model.** Pupil size (PS) is the most influential factor affecting axial elongation rate, with the highest importance score (100), followed by central corneal thickness (CCT, 40.88) and age of onset (17.96). In contrast, anterior chamber depth (ACD), corneal diameter (CD), and corneal curvature (CC, white to wthite, WTW) demonstrated relatively lower predictive contributions.

demonstrated relatively lower predictive contributions. These results indicate that pupil size and corneal thickness serve as critical determinants of axial length changes, providing valuable clinical insights for myopia progression assessment.

SHAP (SHapley Additive exPlanations) values were employed to interpret the machine learning model's predictions, reflecting how much each feature contributed—positively or negatively—to the predicted outcome. The mean absolute SHAP (|SHAP|) value for pupil size was 0.13, with a mean SHAP value of –0.04. For central corneal thickness, the mean |SHAP| value was 0.09, and the mean SHAP value was –0.05. These results indicate that smaller pupil size consistently correlated with faster axial elongation, whereas greater corneal thickness exhibited a protective effect against rapid eye growth. This comprehensive interpretation enhances our understanding of the complex relationships between ocular biometric parameters and myopia development.

## Discussion

This study employed a machine learning-based prediction model for myopia progression utilizing ocular biometric parameters to develop an objective assessment tool for clinical application. The XGBoost algorithm demonstrated superior predictive performance. Pupil size emerged as the most significant determinant of axial elongation rate, followed by corneal thickness and age of onset.

Ocular biometric measurements play a crucial role in predicting myopia progression [11,12]. Our preliminary research has demonstrated a significant correlation between central corneal thickness (CCT) and myopia development, with thinner CCT potentially associated with faster progression in pediatric myopia [7]. The negative correlation observed between CCT and both spherical equivalent progression rate and axial elongation rate suggests that reduced corneal thickness may contribute to accelerated myopic progression [7]. Furthermore, corneal curvature emerges as another critical predictive parameter, as its alterations reflect morphological changes in eyeball structure that influence myopia development. A relevant study utilizing optical coherence tomography revealed significant correlations between retinal curvature, axial length, and mean corneal curvature in myopic patients [13]. Additionally, corneal biomechanical properties, including deformation amplitude and apical radius of curvature, show significant associations with myopia onset and progression [14]. These biomechanical parameter variations likely mirror structural corneal changes during myopia progression, thereby providing valuable reference data for prediction. However, previous studies have been constrained by conventional statistical approaches, failing to establish an effective predictive model for myopia progression. The limitations of traditional analytical methods in capturing complex, multidimensional relationships among ocular parameters have hindered the development of robust prediction systems, highlighting the need for more sophisticated analytical frameworks such as machine learning algorithms.

The application of machine learning techniques in ophthalmology has become increasingly prevalent [15,16], particularly in predicting the onset and progression of myopia through ocular biometric measurements [17,18]. This advanced analytical approach not only enhances prediction accuracy but also provides crucial evidence for early clinical intervention [19–21]. Machine learning models have demonstrated exceptional performance in forecasting myopia development among children and adolescents. A representative study developed five distinct prediction models using random forest, decision tree, extreme gradient boosting, support vector machine, and logistic regression algorithms, which effectively identified key risk factors by analyzing complex interactions between ocular biometric parameters, environmental influences, behavioral patterns, and genetic predispositions, while achieving superior predictive performance in test datasets [17]. Furthermore, machine learning has been successfully applied to predict the risk and progression of high myopia. A multivariate linear regression algorithm developed for this purpose exhibited robust generalizability across both internal and external validation cohorts, offering valuable insights for personalized clinical management of childhood myopia [22]. Longitudinal studies employing machine learning approaches, such as a five-year annual examination of school-aged children using random forest algorithms, have revealed that uncorrected distance visual acuity and spherical equivalent refraction serve as reliable predictors of myopia progression, while parental myopia demonstrates diminishing influence as children age [23]. Collectively, these advancements in machine learning applications have significantly improved prediction accuracy and established scientific foundations for developing personalized intervention strategies. The accumulated evidence provides substantial support for clinical decision-making, enabling more effective prevention and control of myopia development in practice. The integration of machine learning with comprehensive ocular examinations represents a paradigm shift from reactive treatment to proactive prevention in myopia management.

This study developed a machine learning-based predictive model for myopia progression risk utilizing ocular biometric parameters, demonstrating significant advantages over conventional analytical approaches in processing large-scale datasets and identifying complex patterns. The machine learning framework effectively extracted critical predictive factors from extensive biometric measurements, including variables that might be overlooked in traditional statistical analyses, such as axial length, gender, and maternal myopia history, which were identified as significant determinants in previous research [17]. Notably, these models have achieved impressive predictive performance, with one study reporting over 70% accuracy and precision in forecasting myopia progression, highlighting their clinical applicability [17]. Compared to conventional statistical models, machine learning algorithms exhibit superior capability in handling nonlinear relationships and high-dimensional data. While traditional linear regression models may fail to capture intricate associations within ocular biometric data, advanced machine learning techniques including random forests and support vector machines can

effectively model these complex relationships, thereby enhancing prediction accuracy [22,23]. Furthermore, the implementation of cross-validation techniques in machine learning ensures robust internal and external validation of model performance, guaranteeing both reliability and generalizability [3]. Collectively, the application of machine learning in myopia progression risk prediction demonstrates substantial potential for processing complex datasets and improving predictive accuracy. By integrating multiple ocular biometric parameters, these models provide a powerful foundation for developing personalized myopia prevention and control strategies [17,22].

Interestingly, this study revealed that pupil size is the most influential factor affecting the axial elongation rate. Conventional statistical analyses employed in our previous studies failed to identify this association. A prior investigation utilizing a real-world large-scale dataset demonstrated that myopic patients exhibit smaller pupil diameters compared to emmetropic individuals [24]. Notably, their study also verified through XGBoost and random forest machine learning algorithms that the contribution of pupil diameter significantly surpassed other variables [24]. Furthermore, several studies have established that pupil parameters can influence myopia progression [25]. Pupil diameter has also been found to be closely associated with the efficacy of various myopia interventions, including orthokeratology [26] and atropine therapy [27,28]. Regarding the potential mechanism underlying the increased susceptibility to myopia in individuals with smaller pupils, we hypothesize that under identical outdoor lighting conditions, those with constricted pupils may receive reduced effective intraocular light exposure.

This study has several limitations that should be acknowledged. First, the relatively small sample size may introduce instability in the research findings. Furthermore, to maintain consistency across multiple research centers, the study incorporated only selected biometric parameters while excluding potentially important measurements such as vitreous depth and lens thickness, which might compromise the comprehensiveness and accuracy of the conclusions. The research team plans to address these limitations in future investigations to enhance the scientific rigor and reliability of the findings. Moreover, this study identified central corneal thickness (CCT) as a significant predictor of myopia progression; however, the relationship has not yet been quantified. Future research will aim to establish a quantitative model that estimates the corresponding axial length elongation associated with each 1-micron reduction in CCT. Additionally, parameters such as lens thickness may also contribute to the prediction of myopia progression, and we intend to explore this aspect in subsequent studies.

In summary, the machine learning-based prediction model effectively utilizes ocular biometric data to assess myopia progression risk, with pupil size and corneal thickness identified as core predictive factors. This model provides a quantitative tool for early clinical intervention. The integration of machine learning techniques with ocular biometric parameters represents a significant advancement in personalized myopia control approaches, although further validation with expanded datasets and additional parameters is warranted to optimize its predictive performance.

## Supporting information

**S1 File. The raw data of training dataset.**
(CSV)

**S2 File. The raw data of testing dataset.**
(CSV)

**S3 File. The code of R.**
(TXT)

## Author contributions

**Conceptualization:** Peng Zhou, Yan Li.

**Data curation:** Sitong Chen.

**Formal analysis:** Sitong Chen.

**Investigation:** Peng Zhou.

**Methodology:** Sitong Chen.

**Software:** Peng Zhou.

**Supervision:** Yan Li.

**Validation:** Yingli Li.

**Visualization:** Yingli Li.

**Writing – original draft:** Peng Zhou.

**Writing – review & editing:** Peng Zhou, Yan Li.

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
