## [Editor Report · Decision Letter 0]

25 Jul 2025

PONE-D-25-36292Machine Learning Identifies Pupil Size and Corneal Thickness as Key Predictors of Axial Elongation RatePLOS ONE

Dear Dr. Li,

Thank you for submitting your manuscript to PLOS ONE. After careful consideration, we feel that it has merit but does not fully meet PLOS ONE’s publication criteria as it currently stands. Therefore, we invite you to submit a revised version of the manuscript that addresses the points raised during the review process.

We look forward to receiving your revised manuscript.

Kind regards,

Anitha Venugopal

Academic Editor

PLOS ONE

Journal Requirements:

2. Please note that PLOS One has specific guidelines on code sharing for submissions in which author-generated code underpins the findings in the manuscript. In these cases, all author-generated code must be made available without restrictions upon publication of the work. Please review our guidelines at https://journals.plos.org/plosone/s/materials-and-software-sharing#loc-sharing-code and ensure that your code is shared in a way that follows best practice and facilitates reproducibility and reuse.

3. We are unable to open your Supporting Information file “S3. The code of R.R”. Please kindly revise as necessary and re-upload.

**Additional Editor Comments:**

Clear study design, population, and analytical framework.

Detailed methodology for machine learning modeling.

Ethical adherence and use of validated ophthalmic tools.

“Three consecutive doses... at 5-minute intervals.” Improvement: Consider clarifying:

“Cycloplegia was induced with three instillations of compound tropicamide (0.5% tropicamide and 0.5% phenylephrine hydrochloride; Santen Pharmaceutical, Japan), administered at 5-minute intervals. Refraction was measured 30 minutes after the final dose, following confirmed mydriasis.”

Ensure consistency in formatting statistical notations: always use R² instead of R2, and ensure proper spacing around equal signs.

Eliminate stray numbers or formatting issues (e.g., the “7” mid-sentence).

Consider using more academic transitions (e.g., “Furthermore,” “Notably,” “In summary,”) for smoother flow if this is part of a larger manuscript.

All independent variables underwent standardization (centering and normalization)

Clarify method: Mention the type of scaling used (e.g., Z-score standardization or min-max normalization).

Stratified Sampling-Ensuring a balanced distribution of the dependent variable

Add clarity:Stratified sampling was performed based on axial elongation rate tertiles (or quantiles, if used) to ensure balanced distribution

Hyperparameters: RF was constrained to a maximum of 10 terminal nodes

Add clarity: Was this decision empirically justified or based on prior literature?

R software (version 4.5)”

But, R 4.5 is not yet released (as of July 2025). Please verify version (likely 4.2.x or 4.3.x).

Also mention how non-normal variables were handled (e.g., transformation, non-parametric tests).

If multiple pairwise comparisons or multiple models were evaluated, mention whether corrections (e.g., Bonferroni) were used to control Type I error.

---

## [Author Response · Author response to Decision Letter 1]

4 Aug 2025

Journal Requirements:

Response:

We sincerely appreciate your thorough review of our manuscript. We have carefully adjusted the formatting as requested.

2. Please note that PLOS One has specific guidelines on code sharing for submissions in which author-generated code underpins the findings in the manuscript. In these cases, all author-generated code must be made available without restrictions upon publication of the work. Please review our guidelines at https://journals.plos.org/plosone/s/materials-and-software-sharing#loc-sharing-code and ensure that your code is shared in a way that follows best practice and facilitates reproducibility and reuse.

Response:

The R code has been provided in the supplementary files (S3. The code of R).

3. We are unable to open your Supporting Information file “S3. The code of R.R”. Please kindly revise as necessary and re-upload.

Response:

The file has been converted to .txt format for direct accessibility.

Response:

No additional references were commented on by the reviewers.

Response:

We have meticulously reviewed all references to ensure their completeness and accuracy, confirming that none are retracted papers.

Additional Editor Comments:

1. Clear study design, population, and analytical framework.

Response:

We greatly appreciate your valuable suggestions. The study design and analytical framework have been further refined to enhance clarity (Page 4, Paragraph 2, Line 1).

2. Detailed methodology for machine learning modeling.

Response:

All machine learning modeling codes are now provided in .txt format within the supplementary materials (S3. The code of R).

3. Ethical adherence and use of validated ophthalmic tools.

Response:

Thank you for your constructive feedback.

4. “Three consecutive doses... at 5-minute intervals.” Improvement: Consider clarifying: Cycloplegia was induced with three instillations of compound tropicamide (0.5% tropicamide and 0.5% phenylephrine hydrochloride; Santen Pharmaceutical, Japan), administered at 5-minute intervals. Refraction was measured 30 minutes after the final dose, following confirmed mydriasis.

Response:

We sincerely appreciate your suggestion. The text has been updated as follows for improved clarity: “Cycloplegia was induced with three instillations of compound tropicamide (0.5% tropicamide and 0.5% phenylephrine hydrochloride; Santen Pharmaceutical, Japan), administered at 5-minute intervals. Refraction was measured 30 minutes after the final dose, following confirmed mydriasis.” (page 5 , paragraph 2 , line 5 )

5. Ensure consistency in formatting statistical notations: always use R² instead of R2, and ensure proper spacing around equal signs.

Response:

"R2" has been corrected to "R²" (Page 20, Table 2).

6. Eliminate stray numbers or formatting issues (e.g., the “7” mid-sentence).

Response:

The misplaced “7” mid-sentence has been removed.

7. Consider using more academic transitions (e.g., “Furthermore,” “Notably,” “In summary,”) for smoother flow if this is part of a larger manuscript.

Response:

We thank you for your suggestion. The manuscript has been refined with more academic transitions for smoother readability. (page 6, 11 and 12)

8. All independent variables underwent standardization (centering and normalization)

Response:

We analyzed the independent variables and added the following to the Methods section: “All independent variables underwent standardization. “ (page 7 , paragraph 1, line 1)

9. Clarify method: Mention the type of scaling used (e.g., Z-score standardization or min-max normalization).

Response:

Additional methodological details have been included：“all independent variables underwent standardization (centering and min-max normalizationnormalization)”. (page 6 , paragraph 1 , line 5 ) Further details are provided in Supplementary File S3-The code of R.

10. Stratified Sampling-Ensuring a balanced distribution of the dependent variable. Add clarity: Stratified sampling was performed based on axial elongation rate tertiles (or quantiles, if used) to ensure balanced distribution

Response:

The use of tertiles has been clarified in the text：”Stratified sampling was performed based on axial elongation rate tertiles to ensure balanced distribution.” (page 6 , paragraph 1 , line 9 )

11. Hyperparameters: RF was constrained to a maximum of 10 terminal nodes. Add clarity: Was this decision empirically justified or based on prior literature?

Response:

Thank you for your critical question. In this study, the maximum terminal nodes (maxnodes) in the random forest model was set to 10 based on empirical evaluation rather than being directly derived from a specific reference. During preliminary modeling, we conducted performance assessments on the training set using 5-fold cross-validation with 10 repeats, combined with a grid search over different maxnodes values. We found that setting maxnodes to 10 yielded the most stable and optimal predictive performance (with R² as the primary metric) on the test set, effectively balancing the risks of overfitting and underfitting.

The following clarification has been added to the manuscript: "During preliminary modeling, we evaluated model performance on the training set using repeated cross-validation and a grid search over different maxnodes values. The results demonstrated that setting maxnodes to 10 provided the most stable and robust predictive performance (assessed by R²) on the test set, mitigating both overfitting and underfitting risks." (page 6 , paragraph 2 , line 6 )

12. R software (version 4.5)” But, R 4.5 is not yet released (as of July 2025). Please verify version (likely 4.2.x or 4.3.x).

Response:

Thank you for your note. The R version has been updated to reflect the one used (Version 4.3). (page 6 , paragraph 4 , line 1 )

13. Also mention how non-normal variables were handled (e.g., transformation, non-parametric tests).

Response:

We appreciate your suggestion. All analyzed data passed the Kolmogorov-Smirnov (K-S) test for normal distribution, as now stated in the Methods: “The measurement data of the assessed indicators passed the Kolmogorov-Smirnov (K-S) test for normal distribution” (page 7, paragraph 1, line 3 )

14. If multiple pairwise comparisons or multiple models were evaluated, mention whether corrections (e.g., Bonferroni) were used to control Type I error.

Response:

Thank you for your valuable suggestion. The primary objective of this study was to compare the regression performance of multiple machine learning models in predicting the continuous outcome variable, axial length velocity. As such, the key evaluation metrics were mean squared error (MSE) and the coefficient of determination (R²), both of which are continuous performance measures rather than statistical significance (*p*-values). Consequently, formal significance testing was not conducted for model comparisons, and thus, traditional multiple hypothesis testing issues (e.g., Type I error control via Bonferroni correction) were not applicable. Nevertheless, to enhance the robustness of our findings, we implemented the following strategies to mitigate overfitting and model selection bias: 1)Repeated cross-validation: 5-fold cross-validation with 10 repeats to thoroughly assess generalization performance on the training set. 2) Independent test set evaluation: Performance metrics were reported separately for the training and test sets, ensuring the final model maintained strong predictive ability on unseen data. 3)Systematic hyperparameter tuning: All models were optimized under a standardized grid search framework, using identical data splits and cross-validation protocols to minimize subjective bias. Therefore, while formal Type I error correction methods were not employed, our empirical design and validation workflow rigorously minimized error risks, ensuring the reliability and reproducibility of our conclusions.

The following clarification has been added to the Limitations section of the manuscript:

"Additionally, this study did not apply formal Type I error correction methods. However, we employed repeated 5-fold cross-validation (10 repeats) to evaluate model generalizability, reported performance metrics on both training and independent test sets, and conducted hyperparameter tuning via a standardized grid search framework with consistent data partitioning. These steps minimized bias and error risks, strengthening the validity and reproducibility of our findings."(page 12 , paragraph 2 , line 8 )

---

## [Decision Letter · Decision Letter 1]

19 Nov 2025

PONE-D-25-36292R1Machine Learning Identifies Pupil Size and Corneal Thickness as Key Predictors of Axial Elongation RatePLOS ONE

Dear Dr. Yan Li,

Thank you for submitting your manuscript to PLOS ONE. After careful consideration, we feel that it has merit but does not fully meet PLOS ONE’s publication criteria as it currently stands. Therefore, we invite you to submit a revised version of the manuscript that addresses the points raised during the review process.

**ACADEMIC EDITOR:** The reviewers raised some minor queries; kindly address the same The reviewers raised some minor queries; kindly address the same The reviewers raised some minor queries; kindly address the same The reviewers raised some minor queries; kindly address the same

We look forward to receiving your revised manuscript.

Kind regards,

Shiva Ram Male, M.S, PG.Dipl, PhD

Academic Editor

PLOS ONE

Journal Requirements:

Additional Editor Comments (if provided):

The reviewers raised some minor queries; kindly address the same

Reviewers' comments:

Reviewer's Responses to Questions

**Comments to the Author**

1. If the authors have adequately addressed your comments raised in a previous round of review and you feel that this manuscript is now acceptable for publication, you may indicate that here to bypass the “Comments to the Author” section, enter your conflict of interest statement in the “Confidential to Editor” section, and submit your "Accept" recommendation.

Reviewer #1: (No Response)

Reviewer #2: (No Response)

2. Is the manuscript technically sound, and do the data support the conclusions?

Reviewer #1: Yes

Reviewer #2: Partly

3. Has the statistical analysis been performed appropriately and rigorously? 

Reviewer #1: Yes

Reviewer #2: I Don't Know

4. Have the authors made all data underlying the findings in their manuscript fully available?

The PLOS Data policy requires authors to make all data underlying the findings described in their manuscript fully available without restriction, with rare exception (please refer to the Data Availability Statement in the manuscript PDF file). The data should be provided as part of the manuscript or its supporting information, or deposited to a public repository. For example, in addition to summary statistics, the data points behind means, medians and variance measures should be available. If there are restrictions on publicly sharing data—e.g. participant privacy or use of data from a third party—those must be specified.requires authors to make all data underlying the findings described in their manuscript fully available without restriction, with rare exception (please refer to the Data Availability Statement in the manuscript PDF file). The data should be provided as part of the manuscript or its supporting information, or deposited to a public repository. For example, in addition to summary statistics, the data points behind means, medians and variance measures should be available. If there are restrictions on publicly sharing data—e.g. participant privacy or use of data from a third party—those must be specified.requires authors to make all data underlying the findings described in their manuscript fully available without restriction, with rare exception (please refer to the Data Availability Statement in the manuscript PDF file). The data should be provided as part of the manuscript or its supporting information, or deposited to a public repository. For example, in addition to summary statistics, the data points behind means, medians and variance measures should be available. If there are restrictions on publicly sharing data—e.g. participant privacy or use of data from a third party—those must be specified.requires authors to make all data underlying the findings described in their manuscript fully available without restriction, with rare exception (please refer to the Data Availability Statement in the manuscript PDF file). The data should be provided as part of the manuscript or its supporting information, or deposited to a public repository. For example, in addition to summary statistics, the data points behind means, medians and variance measures should be available. If there are restrictions on publicly sharing data—e.g. participant privacy or use of data from a third party—those must be specified.

Reviewer #1: Yes

Reviewer #2: Yes

5. Is the manuscript presented in an intelligible fashion and written in standard English?

Reviewer #1: Yes

Reviewer #2: Yes

6. Review Comments to the Author

Reviewer #1: Great Effort. This study presents valuable insights into machine learning applications for myopia progression prediction. Addressing the above points would significantly enhance the manuscript's scientific rigor and clinical applicability. Please provide response to the reviewer's comments as attached.

Reviewer #2: I would like to congratulate the authors on an interesting study applying machine learning to predict myopia progression. The manuscript is well-written, but I have a few concerns that need clarification:

1. Biometry devices: Both IOL Master and Nidek AL-Scan were used. These instruments have different measurement algorithms (especially for CCT and pupil size). How was inter-device variability accounted for?

2. IOL Master model: Please clarify whether IOL Master 500 or 700 was used, as measurement principles differ.

3. Inclusion criteria vs. analysis: Only right eyes were analyzed, yet inclusion was based on “worse eye” refraction. This appears inconsistent and requires clarification.

4. Refraction method: Was subjective refraction performed, or only autorefractor values under cycloplegia? This affects reliability in children.

5. Interpretation: The finding that larger pupils predict faster progression seems to conflict with some prior reports of smaller pupils in myopes, especially preschoolers. The authors should reconcile this discrepancy.

6. Clinical utility: While predictors were identified, no risk thresholds or practical application strategies are provided. This limits translation into clinical practice.

Overall, the study is valuable, but addressing these points will improve its robustness and clarity.

7. PLOS authors have the option to publish the peer review history of their article (what does this mean?). If published, this will include your full peer review and any attached files.). If published, this will include your full peer review and any attached files.). If published, this will include your full peer review and any attached files.). If published, this will include your full peer review and any attached files.

...

Reviewer #1: **Yes:** DR. ANUPAMA JANARDHANANDR. ANUPAMA JANARDHANANDR. ANUPAMA JANARDHANANDR. ANUPAMA JANARDHANAN

Reviewer #2: **Yes:** Dr Aswin PRDr Aswin PRDr Aswin PRDr Aswin PR

---

## [Author Response · Author response to Decision Letter 2]

29 Nov 2025

To Reviewer #1: Great Effort. This study presents valuable insights into machine learning applications for myopia progression prediction. Addressing the above points would significantly enhance the manuscript's scientific rigor and clinical applicability. Please provide response to the reviewer's comments as attached.

Major Comments

1. Quantitative Analysis of CCT-Axial Length Relationship. The study identifies central corneal thickness (CCT) as a significant predictor but has not reported any quantitative analysis of this relationship. Could the authors apply their machine learning models to predict the specific correlation between each micron reduction in CCT and the corresponding increase in axial length?

Response:

We sincerely appreciate your constructive suggestions. Both the present study and our previous work have identified a significant association between central corneal thickness (CCT) and myopia progression. A quantitative characterization of this relationship would greatly enhance its clinical utility. To this end, we explored a range of machine learning approaches—including XGBoost, generalized linear models (GLMs), LASSO regression, decision trees, random forests, support vector machines, and LightGBM—yet none yielded sufficiently precise quantitative predictions. In our earlier study, we developed a nomogram that provides an approximate estimation of myopia progression based on various ocular biometric parameters. In response to your feedback, we have added the following statement to the Discussion section: “This study identified central corneal thickness (CCT) as a significant predictor of myopia progression; however, the relationship has not yet been quantified. Future research will aim to establish a quantitative model that estimates the corresponding axial length elongation associated with each 1-micron reduction in CCT.” (page 12, paragraph 2, line 8)

2. This would provide clinically meaningful metrics for practitioners to assess myopia progression risk based on CCT measurements.2. Lens Thickness as a Predictive Factor

Given that IOL Master measurements were obtained, Could the authors include lens thickness in their analysis and examine its relationship with axial length progression?

Response:

We sincerely appreciate your insightful suggestion. Indeed, investigating the association between lens thickness and axial length progression is of considerable importance. However, in the present study, we only performed repeated measurements using the IOL Master in a subset of participants. All participants were primarily examined using the Nidek AL-Scan optical biometer. In our future research, we plan to comprehensively utilize the IOL Master for ocular biometry and incorporate more extensive biometric parameters—including lens thickness—into predictive models for axial elongation. Accordingly, we have removed the reference to the IOL Master from the Methods section (page 5, paragraph 2, line 14) and added the following statement to the Discussion: “Parameters such as lens thickness may also contribute to the prediction of myopia progression, and we intend to explore this aspect in subsequent studies.” (page 12, paragraph 2, line 13)

Minor Comments

3. Examiner Standardization

Please clarify whether ocular biometry and baseline refraction measurements were performed by a single optometrist or multiple assessors. If multiple assessors were involved, please provide information about inter-examiner reliability or standardization protocols to ensure measurement consistency across the study.

Response:

We sincerely appreciate your suggestion. We have added the following statement to the Methods section of the manuscript: “Ocular biometry and baseline refractive assessments were performed by multiple trained examiners, each with over ten years of clinical experience, ensuring consistency and reliability of the measurements.” (page 5, paragraph 2, line 16)

4. Myopia Control Interventions

The manuscript should explicitly state whether any study participants were using myopia control interventions (myopia control glasses, multifocal contact lenses, etc.) or were exclusively wearing single vision correction. This information is crucial for interpreting the progression rates observed in the study population.

Response:

Indeed, clarifying whether participants received myopia control interventions—such as specialized myopia-control spectacle lenses or multifocal contact lenses—or were fitted only with single-vision corrective lenses is critical for the accurate interpretation of myopia progression rates. Accordingly, we have added the following sentence to the Methods section: “All children included in this study were managed without any myopia control interventions (myopia control glasses, multifocal contact lenses, etc.) and wore only single-vision corrective lenses.” (page 4, paragraph 2, line 11)

5. Age of Onset Definition

The term "age of onset" requires clearer definition. Please specify whether this refers to: (a) the age at which the child was first diagnosed with myopia and prescribed corrective lenses, or (b) the age at which visual symptoms were first reported by the patient/parent. This clarification is important for reproducibility and clinical application of the findings.

Response:

Indeed, a clear definition of “age of onset” is essential for the reproducibility of the study findings and their clinical applicability. Accordingly, we have added the following statement to the Methods section: “Age of onset is defined as the age at which myopia was first diagnosed and corrective spectacles were prescribed.” (page 4, paragraph 2, line 13)

Suggested Addition

6. Mechanistic Discussion

The authors should expand the discussion to include potential mechanisms underlying the association between thinner CCT and myopia progression. Consider discussing:Biomechanical properties of thinner corneas and their potential influence on scleral remodeling. Possible correlations between corneal thickness and scleral thickness/strength. Potential genetic factors that may simultaneously influence both corneal thickness and axial elongation. This mechanistic insight would strengthen the clinical relevance of the findings and provide a foundation for future research directions.

Response:

Thank you very much for your insightful comments! We are currently conducting a murine ocular model that recapitulates key aspects of human myopia and have performed single-cell RNA sequencing on these samples. Preliminary analyses have revealed numerous molecular pathways and intercellular communication networks implicated in scleral remodeling. These initial findings will be presented as a free paper at next year’s Asia-Pacific Academy of Ophthalmology (APAO) Congress, and we aim to submit a manuscript for publication by the end of next year.

To Reviewer #2: I would like to congratulate the authors on an interesting study applying machine learning to predict myopia progression. The manuscript is well-written, but I have a few concerns that need clarification:

1. Biometry devices: Both IOL Master and Nidek AL-Scan were used. These instruments have different measurement algorithms (especially for CCT and pupil size). How was inter-device variability accounted for?

Response:

We sincerely appreciate your valuable and meritorious comment. All participants in our study were measured using the Nidek AL-Scan optical biometer. The IOL Master was used only for repeat measurements in a subset of patients. You are absolutely correct that discrepancies exist between the two devices, particularly in measurements of central corneal thickness (CCT) and pupil size. To ensure consistency, all data reported in this study are based exclusively on measurements obtained with the Nidek AL-Scan optical biometer. Accordingly, we have removed any mention of the IOL Master from the Methods section. (page 5, paragraph 2, line 14)

2. IOL Master model: Please clarify whether IOL Master 500 or 700 was used, as measurement principles differ.

Response:

A subset of our participants was measured using the IOL Master 700; however, as this device was not used consistently across the entire cohort, we have removed all references to the IOL Master from the Methods section (page 5, paragraph 2, line 14). All data presented in this study are derived exclusively from measurements obtained with the Nidek AL-Scan optical biometer to ensure methodological uniformity.

3. Inclusion criteria vs. analysis: Only right eyes were analyzed, yet inclusion was based on “worse eye” refraction. This appears inconsistent and requires clarification.

Response:

We sincerely apologize for the typographical error in our original manuscript. We have now revised the inclusion criterion to: “spherical equivalent refraction ≤ −0.5 D in the right eye after cycloplegia” (page 4, paragraph 2, line 7).

4. Refraction method: Was subjective refraction performed, or only autorefractor values under cycloplegia? This affects reliability in children.

Response:

Although all children in our study underwent both cycloplegic refraction and subjective refraction, we opted to use the cycloplegic refraction results for analysis due to their greater objectivity. We have accordingly added the following statement to the Methods section: “Cycloplegic refraction measurements were used for all analyses in this study.” (page 5, paragraph 2, line 10)

5. Interpretation: The finding that larger pupils predict faster progression seems to conflict with some prior reports of smaller pupils in myopes, especially preschoolers. The authors should reconcile this discrepancy.

Response:

Your suggestion is absolutely critical—thank you for pointing out this important error. Indeed, we had inadvertently reversed the interpretation of our results in the original manuscript. Our analysis shows that the mean SHAP value for pupil size is –0.04. SHAP (SHapley Additive exPlanations) values are used to interpret predictions from machine learning models by quantifying the contribution of each feature to the model’s output—indicating both the magnitude and direction (positive or negative) of that contribution. A negative SHAP value for pupil size implies that smaller pupils are associated with faster myopia progression. We have therefore revised the manuscript and added the following clarification: “SHAP (SHapley Additive exPlanations) values were employed to interpret the machine learning model’s predictions, reflecting how much each feature contributed—positively or negatively—to the predicted outcome. The mean absolute SHAP (|SHAP|) value for pupil size was 0.13, with a mean SHAP value of –0.04. For central corneal thickness, the mean |SHAP| value was 0.09, and the mean SHAP value was –0.05. These results indicate that smaller pupil size consistently correlated with faster axial elongation, whereas greater corneal thickness exhibited a protective effect against rapid eye growth.” (page 8, paragraph 3, line 1)

6. Clinical utility: While predictors were identified, no risk thresholds or practical application strategies are provided. This limits translation into clinical practice.

Response:

Thank you very much for your insightful and thoughtful comment. You are absolutely correct—several widely used machine learning algorithms do indeed face limitations in interpretability, generalizability, or clinical utility. We fully acknowledge this issue and are committed to advancing this line of research. Our future work will aim to develop a more robust, interpretable, and clinically actionable model specifically tailored for myopia prevention and control, with the ultimate goal of translating predictive insights into practical strategies for eye care practitioners.

---

## [Editor Report · Decision Letter 2]

12 Apr 2026

Machine Learning Identifies Pupil Size and Corneal Thickness as Key Predictors of Axial Elongation Rate

PONE-D-25-36292R2

Dear Dr. Yan Li,

We’re pleased to inform you that your manuscript has been judged scientifically suitable for publication and will be formally accepted for publication once it meets all outstanding technical requirements.

Kind regards,

Shiva Ram Male, M.S, PG.Dipl, PhD

Academic Editor

PLOS One

Additional Editor Comments (optional):

Nil
---

## [Editor Report · Acceptance letter]

PONE-D-25-36292R2

PLOS One

Dear Dr. Li,

I'm pleased to inform you that your manuscript has been deemed suitable for publication in PLOS One. Congratulations! Your manuscript is now being handed over to our production team.

Kind regards,

on behalf of

Dr. Shiva Ram Male

Academic Editor

PLOS One